# An Innovative Approach to Alleviate Zinc Oxide Nanoparticle Stress on Wheat through Nanobubble Irrigation

**DOI:** 10.3390/ijms25031896

**Published:** 2024-02-05

**Authors:** Feng Zhang, Shuxin Li, Lichun Wang, Xiangnan Li

**Affiliations:** 1Key Laboratory of Black Soil Conservation and Utilization, Northeast Institute of Geography and Agroecology, Chinese Academy of Sciences, Changchun 130102, China; fe_stu@outlook.com (F.Z.); wsylsx2015@outlook.com (S.L.); 2College of Advanced Agricultural Sciences, University of Chinese Academy of Sciences, Beijing 100049, China; 3Key Laboratory of Crop Eco-Physiology and Farming System in the Northeastern, Institute of Agricultural Resources and Environment, Ministry of Agriculture and Rural Affair, Jilin Academy of Agricultural Sciences, Changchun 130033, China

**Keywords:** zinc oxide nanoparticles, nanobubble irrigation, plant growth, soil physicochemical property, nutrient limitation, photosynthesis, OJIP curve

## Abstract

The extensive utilization of zinc oxide nanoparticles in consumer products and the industry has led to their substantial entry into the soil through air and surface runoff transportation, which causes ecotoxicity in agro-ecosystems and detrimental effects on crop production. Nanobubbles (diameter size < 1 µm) have many advantages, such as a high surface area, rapid mass transfer, and long retention time. In this study, wheat seedlings were irrigated with a 500 mg L^−1^ zinc oxide nanoparticle solution delivered in the form of nanobubble watering (nanobubble-ZnO-NPs). We found that nanobubble watering improved the growth and nutrient status of wheat exposed to zinc oxide nanoparticles, as evidenced by increased total foliar nitrogen and phosphorus, along with enhanced leaf dry mass per area. This effect can be attributed to nanobubbles disassembling zinc oxide aggregates formed due to soil organic carbon, thereby mitigating nutrient absorption limitations in plants. Furthermore, nanobubbles improved the capability of soil oxygen input, leading to increased root activity and glycolysis efficiency in wheat roots. This work provides valuable insights into the influence of nanobubble watering on soil quality and crop production and offers an innovative approach for agricultural irrigation that enhances the effectiveness and efficiency of water application.

## 1. Introduction

Nanomaterials are used in diverse fields, such as biomedical industries, agriculture, and cosmetics [1,2]. Zinc oxide nanoparticles (ZnO-NPs), as one of the most extensively used types of nanomaterials, have a huge annual capacity, with production ranging from 550 to 35,000 metric tons per year in different countries [3]. However, the increasing usage of ZnO-NPs leads to serious environmental emissions [4]. These ZnO-NPs inevitably enter the agro-ecosystem through various approaches, such as industrial wastewater, domestic sewage, and fertilizers [5,6,7]. The persistence of ZnO-NPs in agro-ecosystems has been observed, and their concentration is predicted to steadily increase in the soil. This poses a substantial potential threat to soil ecosystem functions and plant growth, thereby risking human dietary health through the food chain [4,8,9]. A high concentration of ZnO-NPs changes the physicochemical properties of soil, such as increasing soil pH, decreasing soil redox potential (*Eh*), and total oxygen demand [10,11]. Furthermore, ZnO-NPs also change the soil microbial communities and decrease the activity of soil enzymes, hence affecting the nutrient cycles. For example, ZnO-NPs significantly decrease the bacterial and fungal community diversity and biomass and inhibit the activities of catalase (CAT), phosphatase, peroxidase, and β-glucosidase, affecting soil carbon and nitrogen mineralization [12,13,14,15,16]. Previous studies have demonstrated that ZnO-NPs adhering to root surfaces can be absorbed into the roots and subsequently transported into different plant tissues. The absorbed ZnO-NPs inhibit root elongation and nutrition absorption in various plant species, alter the physiological processes in roots, and reduce biomass [9,17,18]. For various plants, such as wheat (*Triticum aestivum* L.), rice (*Oryza sativa* L.), and maize (*Zea mays* L.), it has been shown that ZnO-NPs adversely affect both crop root elongation and biomass production [19]. Also, it has been reported that high concentrations of ZnO-NPs concededly hinder nitrogen (N) and phosphorus (P) acquisition and impair photosynthesis in maize [20]. Meanwhile, our previous study showed that ZnO-NPs significantly induce reactive oxygen species (ROS) accumulation and simultaneously regulate antioxidant- and carbohydrate-related enzyme activities by protein phosphorylation, thereby inducing oxidative damage in *Hordeum vulgare* L. [21]. In summary, the excessive presence of ZnO-NPs in agro-systems can impose various adverse effects on crops. Nevertheless, strategies to alleviate the ZnO-NPs toxicity to ecosystems, particularly agro-systems, remain largely unclear [6]. It is known that the aggregation, immobilization, and dissolution that are intricately linked to the toxicity of ZnO-NPs can be influenced by key soil properties, such as pH, *Eh*, and soil organic carbon (SOC). For example, high soil pH and organic matter inversely correlate with Zn^2+^ released from ZnO-NPs, resulting in decreased transformations and phytotoxicity [22,23].

Nanobubbles (diameter size < 1 µm) were characterized by a high surface area, slow rising velocity in water, quickness in mass transfer, long retention time, and the generation of free radicals [24,25]. Due to these unique characteristics, nanobubbles have been applied in agriculture. Previous studies have revealed that irrigation with nanobubble-containing water affects the microbial communities, improves the activity of soil extracellular enzymes, increases soil fertility and available nutrients, and promotes nutrition uptake, subsequently enhancing plant performance [26,27,28]. In addition, nanobubbles have shown the potential to mitigate the toxicity of heavy metals to plants, and the continuous retention time allows the nanobubbles to have a long-term effect on the soil [29,30]. Recent research showed that nanobubbles with oxygen reduce the absorption of arsenic by affecting the rhizosphere *Eh* and dissolved oxygen (Do), which reduces arsenic bioavailability and improves root activity and crop growth in rice [31,32]. Meanwhile, some studies indicated that nanobubbles with hydrogen or/and oxygen can increase photosynthetic pigment contents, promote photosynthesis, and improve the activity of antioxidant enzymes, thereby alleviating cadmium stress in crop plants [30,33]. Moreover, the nanobubble has been indicated to affect the soil characteristics, which may be associated with the influence on the bioavailability and ecotoxicity of ZnO-NPs in plants [22,34]. 

To elucidate the physiological toxicity of ZnO-NPs on crops and investigate potential mechanisms for alleviating this toxicity using nanobubbles, we conducted nanobubble irrigation experiments on wheat (*Triticum aestivum* L.) exposed to ZnO-NPs. We assessed the photosynthetic performance and metabolic processes of wheat under different treatment conditions. In addition, we measured soil porosities, including *Eh*, redox-active substances, and enzyme activities, to understand the plant-soil interactions. We hypothesized that (1) ZnO-NPs induce ecotoxicity in wheat. (2) However, this toxicity was effectively mitigated by irrigation with nanobubble-containing water, possibly because it changed the soil *Eh* and enzyme activity and constrained the ZnO-NPs transformations. (3) Moreover, the nanobubble watering enhanced the photosynthesis and antioxidant capacity, which also contributed to the improved performance of wheat under ZnO-NP stress.

## 2. Results

### 2.1. Zn Concentration and Plant Performance

The soil Zn concentration exhibited a significant 48.18% increase in response to irrigation with the conventional ZnO-NPs solution (Figure 1A). Intriguingly, the combination of nanobubbles and ZnO-NPs resulted in a 2.94% reduction in soil Zn concentration compared to the conventional method. Zn concentration in root showed an approximately 2.7-fold increase with the conventional ZnO-NPs solution, a trend further exacerbated in the nanobubble-ZnO-NPs irrigation group. Additionally, conventional ZnO-NPs irrigation had a detrimental effect on shoot fresh and dry weights but caused an increase in plant height (Figure 1B,C). Interestingly, nanobubble-ZnO-NP irrigation exhibited a reversal of these trends, with higher shoot fresh and dry weights but lower plant height in comparison to ZnO-NP irrigation.

### 2.2. Water Quality Properties, Soil Physicochemical Characteristics, and Enzymatic Properties

The dissolved oxygen (Do) concentration exhibited noticeable variations under different irrigation conditions (Figure 2A). The highest concentration (8.084 mg L^−1^) was observed under nanobubble irrigation without ZnO-NPs (ZnO_0_nW treatment), followed by nanobubble irrigation with ZnO-NPs (ZnO_500_nW treatment). The lowest Do concentration (5.818 mg L^−1^) was from normal water irrigation mixed with 500 mg L^−1^ ZnO-NPs (ZnO_500_W treatment). In terms of physicochemical characteristics, conventional ZnO-NPs irrigation significantly increased soil pH, *Eh*, the concentration of SOC, and soil dissolved organic carbon (DOC) (Figure 2B,F,G,M). Conversely, the concentrations of soil water-soluble phenol and ferrous (Fe (II)), as well as the ratio of ferrous to ferric iron (Fe(II)/Fe(III)), were significantly decreased by conventional ZnO-NPs irrigation in comparison to pure water irrigation (Figure 2C–E). Similarly, nanobubble-ZnO-NPs irrigation caused dramatic increases in *Eh* and concentrations of water-soluble phenol, DOC, and SOC in soil compared to conventional ZnO-NPs irrigation (Figure 2E,F,G,M).

In addition to influencing physicochemical characteristics, conventional ZnO-NPs irrigation also affected soil enzyme activities, as manifested by the significant decreases in the activities of soil CAT and β-1,4-glucosidase (BG), as well as the significant increases in the activities of L-leucine aminopeptidase (LAP) and neutral phosphatase (NP) (Figure 2H,I; Appendix A). Moreover, the enzymatic ratios of C:N ratio and C:P ratio were significantly reduced under conventional ZnO-NPs irrigation in comparison to pure water irrigation (Figure 2J–L). In contrast to the effects of conventional ZnO-NPs irrigation, nanobubble-ZnO-NPs irrigation enhanced the activities of soil enzymes CAT and BG and accordingly increased enzymatic C:N and C:P ratios, while LAP activity was significantly reduced.

### 2.3. Plant Stress Damage and Antioxidant Enzyme Activities

To identify the potential risk for plants caused by ZnO-NPs, we investigated the root activity and ROS in roots and leaves. We observed that root activity was significantly decreased by 49.82% in wheat seedlings with conventional ZnO-NPs irrigation compared with that with water irrigation (Figure 3A). Compared with the control, an increase trend was found in roots with conventional ZnO-NPs irrigation, showing *c*. 1.50- and 1.56-fold higher concentrations of H_2_O_2_ and O_2_^−^ (Figure 3B,C). Meanwhile, the malondialdehyde (MDA) concentration was slightly increased by the conventional ZnO-NPs irrigation as compared to the pure water irrigation (Figure 3D). The nanobubble-treated ZnO-NPs irrigation caused a significant increase in root activity and O_2_^−^ concentration in roots but a significant decrease in concentrations of H_2_O_2_ and MDA compared with the conventional ZnO-NPs irrigation (Figure 3B–D). In leaves, the conventional ZnO-NPs irrigation significantly decreased the O_2_^−^ concentration while significantly increasing the MDA concentration (Figure 3B,D). By contrast, the decrease in O_2_^−^ concentration was to a much lesser extent in leaves with the nanobubble-treated ZnO-NPs irrigation (Figure 3C). When wheat plants were irrigated with the conventional ZnO-NPs solution, some antioxidant enzyme activities showed a significant downtrend, including the activities of ascorbate peroxidase (APX) and glutathione S-transferase (GST) in roots, as well as GST activity in leaves; however, the activities of glutathione reductase (GR), monodehydroascorbate reductase (MDHAR), and superoxide dismutase (SOD) were significantly increased in leaves (Figure 3E,F; Appendix A). Interestingly, in comparison with conventional ZnO-NPs irrigation, the activities of some more antioxidant enzymes were further declined in roots with nanobubble-treated ZnO-NPs irrigation, including APX, dehydroascorbate reductase (DHAR), GR, and GST. Meanwhile, the activities of two antioxidant enzymes, namely DHAR and MDHAR, were also declined, while GR activity was raised in leaves with nanobubble-treated ZnO-NPs irrigation.

### 2.4. Glucose Concentration and Key Carbohydrate Metabolism Enzyme Activities

To explore how different forms of ZnO-NP irrigation affect energy metabolism, the glycolysis and pentose phosphate pathways were tested. The glucose concentrations were significantly decreased by 56.56% and 20.97% by the conventional ZnO-NPs irrigation in roots and leaves, compared with the control (Figure 4A). In relation to conventional ZnO-NPs irrigation, nanobubble-treated ZnO-NPs irrigation remitted the decrease in glucose concentration, as expressed by 8.93% and 55.66% increases in glucose concentrations in roots and leaves, respectively. We further tested the enzyme activities related to glycolysis and the pentose-phosphate pathway. Under conventional ZnO-NPs irrigation, increasing trends were found for the activities of ADP-Glucose pyrophosphorylase (AGPase), UDP-Glucose phosphorylase (UGPase), and cell wall invertase (cwInv) in roots, as well as the phosphofructokinase (PFK) activity in leaves, while a decreasing trend was observed for the activities of PFK, phosphoglucomutase (PGM), cytoplasmic invertase (cytInv), and vacuolar invertase (vacInv) in roots, as well as the activities of AGPase, fructokinase (FK), hexokinase (HXK), phosphoglucoisomerase (PGI), UGPase, cwInv, and vacInv in leaves (Figure 4B,C). Comparing the nanobubble-treated ZnO-NPs with conventional ZnO-NPs irrigation, the nanobubble-treated ZnO-NPs irrigation significantly enhanced the activities of glucose-6-phosphate dehydrogenase (G6PDH) and PFK in roots and the activities of FK, UGPase, cytInv, and vacInv in leaves. Also, it reduced the UGPase activity in roots and the activities of AGPase, aldolase (Ald), PFK, and PGI in leaves as compared to conventional ZnO-NPs irrigation.

### 2.5. Trade-off Strategies for Leaf Functional Traits

A quantitative analysis of leaf functional traits was performed to elucidate the changes in wheat stress tolerance. The conventional ZnO-NPs irrigation significantly reduced the leaf dry mass per area (LMA) and *N*_area_ by 16.57% and 12.06% in relation to the control but significantly enhanced the *A*_area_ by 12.47% (Figure 5A,B,D). Notably, *R*_dark_, area, and *P*_area_ appeared slightly declined (Figure 5C,E). Meanwhile, leaf thickness was significantly increased by 26.62% when plants were subjected to conventional ZnO-NPs irrigation (Figure 5F). Further inspection of the leaf economics spectrum (LES) demonstrated that the nanobubble-treated ZnO-NPs irrigation induced 76.30%, 89.03%, and 68.42% higher increases in LMA, Narea, and *P*_area_ than the conventional ZnO-NPs irrigation, accompanied by slight increases in *A*_area_ and *R*_dark_, area.

### 2.6. Photosynthetic Performances

Considering the significant ecological differences revealed by LES, we examined photosynthesis in detail, which is the most pivotal eco-physiological function of the leaf. The photosynthetic pigment concentrations, including chlorophyll and carotenoid, were not significantly changed by the conventional ZnO-NPs irrigation (Figure 6A). However, the nanobubble-treated ZnO-NPs irrigation increased the total chlorophyll concentration by 13.08% compared to conventional ZnO-NPs irrigation. For gas exchange parameters and biochemical capacities, the increases were found under conventional ZnO-NPs irrigation, with 40.04% and 21.92% increases in net photosynthetic rate (*A*_n_) and maximum electron transport rate (*J*_max_), respectively; however, an 11.52% decrease was found in intercellular CO_2_ concentration (*C*_i_) (Figure 6A,E,G). In addition, the slight increases in *T*_r_ ang gs were caused by the conventional ZnO-NPs irrigation in relation to the control (Figure 6C,D). As compared to the conventional ZnO-NPs irrigation, only Jmax was dramatically decreased by 19.60% under the nanobubble-treated ZnO-NPs irrigation. Meanwhile, slight decreases in *T*_r_ ang *g*_s_ were found under the nanobubble-treated ZnO-NPs irrigation in relation to the conventional ZnO-NPs irrigation.

From the photosynthetic system II perspective, we found that the conventional ZnO-NPs irrigation only caused a significant decline in dissipated energy flux per RC (DI_O_/RC) and a slight decline in performance index (*PI*_abs_), while the nanobubble-treated ZnO-NPs irrigation enhanced the *PI*_abs_ compared with the conventional ZnO-NPs irrigation (Figure 6H,I). In addition to the above parameters, we also paid attention to the photosynthetic nitrogen use efficiency (PNUE) and N allocation within the photosynthetic apparatus. The conventional ZnO-NPs irrigation induced 27.86%, 37.73%, and 21.93% increases in PNUE, the fraction of nitrogen allocated to electron transport components (*PN*_et_), and the fraction of nitrogen allocated to the carboxylation system (*PN*_cb_), accordingly decreasing the fraction of nitrogen allocated to non-photosynthetic nitrogen (*PN*_non-psn_) by 14.83% (Figure 6K) in relation to the control. As compared to conventional ZnO-NPs irrigation, the nanobubble-treated ZnO-NPs irrigation caused significant reductions in PNUE, *PN*_cb_, and *PN*_et_, while induced a significant increase in *PN*_non-psn_.

### 2.7. Water Relations

From the water relations perspective, water-use efficiency, including intrinsic water-use efficiency (WUE_i_) and instantaneous water-use efficiency (WUE_inst_), was significantly enhanced when the wheat seedlings were irrigated with the conventional ZnO-NPs solution, wherein the nanobubble-treated ZnO-NPs irrigation did not play key roles in water-use efficiency (Figure 7A). Otherwise, as compared to the water irrigation control, the water potential was significantly lowered under the conventional ZnO-NPs in leaves (Figure 7B). The nanobubble-treated ZnO-NPs irrigation dramatically enhanced water potential in leaves compared with conventional ZnO-NPs irrigation.

## 3. Discussion

The biotoxicity of ZnO-NPs has attracted increasing attention due to their extensive application and subsequently unregulated/unsafe release, although their benefits can be seen [19,35]. The accumulation of ZnO-NPs, such as polluted water irrigation (mainly from industrial plants and domestic greywater), inducing ecotoxicity in agro-ecosystems, especially crop production, can directly impact agricultural sustainability and food safety, which warrants major attention [6]. However, we have not obtained a holistic understanding of the soil-plant system responses to ZnO-NPs. Otherwise, it has been reported thus far that nanobubbles, defined as bubbles with a diameter < 1 µm, affect the soil characteristics and crop absorption for macro- and micro-elements such as selenium, N, and P [25,36,37]. However, it is not yet well understood how the nanobubbles affect the ecotoxicity of nanoparticulates, including ZnO-NPs, and what the mechanisms are underlying this effect.

### 3.1. Nanobubble Treatment Enhances Wheat Plant Performance

Our results showed that shoot dry and fresh weights were decreased by conventional ZnO-NPs irrigation, suggesting that the ZnO-NPs caused stress effects on plant growth (Figure 1B). This result was consistent with the previous study [38]. Meanwhile, many studies indicated that ZnO-NPs increased the plant weight, which is in line with our results (Figure 1C) [39]. Conversely, the nanobubble treatment increased the shoot dry and fresh weights but decreased the plant height, suggesting that the nanobubble treatment contributed to promoting plant growth, thereby enhancing the ZnO-NPs resistance. Huang et al. also found that nanobubble treatment mitigates cadmium (Cd) stress and increases the biomass in rice [29]. 

### 3.2. Nanobubble Treatment May Regulate the Ecotoxicity of ZnO-NPs via Changing the Eh and SOC Accumulation

Soil, as a kind of complex system, can provide the medium for plant growth, and the biochemical, physical, and chemical reactions in soil are prone to being disturbed by nanoparticulate input, thereby causing changes in soil properties [40]. The previous study demonstrated that metal oxide nanoparticle input, including ZnO-NPs and CuO-NPs, increased the soil *Eh* and pH [11]. Similarly, here, conventional ZnO-NPs irrigation caused the enhancement of soil *Eh*, and we considered why (Figure 2G). On one hand, we surmised that ZnO-NPs per se possessed the charged surfaces formed by the hydroxyl (-OH) group, thereby enhancing *Eh* [19]. On the other hand, in soil, the water-soluble phenol could promote the abiotic transformation of oxygen to O_2_^−^; the dismutation and hydrolysis related to O_2_^−^ were next proceeded to produce H_2_O_2_; the H_2_O_2_ was further used to produce the -OH through the Fe(II)-mediated Fenton reactions [41]. Interestingly, the same trend was found in redox substances in our results, by which the concentrations of water-soluble phenol and Fe(II) and Fe(II)/Fe(III) were decreased by the conventional ZnO-NPs irrigation, suggesting that more -OH was produced by consuming more water-soluble phenol and Fe(II) (Figure 2C,D). Furthermore, the decreased CAT activity, which is responsible for the conversion of H_2_O_2_ to H_2_O and O_2_, also indicated the potential for an increase in -OH (Figure 2H). Due to the change in Fe(II), we investigated the SOC sequestration mediated by the Fe(II)/Fe(III) cycle, which was a critical avenue [42]. As previously described, the conventional ZnO-NPs irrigation lowered Fe(II) concentration and Fe(II)/Fe(III), accompanied by an increase in DOC, suggesting the formation of Fe-bound organic carbon may be increased (Figure 2F) [43]. This was consistent with the increase in SOC under conventional ZnO-NPs irrigation (Figure 2M). Contradictorily, the DOC still maintained a relatively higher level, possibly attributed to the input of root exudates stimulated by ZnO-NPs [44]. Another important aspect was that the SOC could increase the ZnO-NPs aggregation through organic matter adsorption, thereby causing adhesion to the root surface [6,45,46]. This raised the possibility that the absorption of nutrient elements was hindered due to the existence of an adhesion layer. The increased Zn concentrations in soil and roots provided support for the theoretical explanation developed above (Figure 1A). The previous conclusion showed that soil pH was negatively correlated with the level of available Zn from ZnO-NPs. In our results, a higher soil pH, representing the decreased available Zn, also in turn implied that the ZnO-NPs aggregation was aggravated (Figure 2B) [19]. Furthermore, the Zn dissolution could also be a reason [14]. The conventional ZnO-NPs irrigation not only affects soil physicochemical properties but also regulates microbial activities [47]. The enzymatic C:N ratio and C:P ratio resulted from decreased C-acquiring enzyme activity (i.e., BG), and the activities of N- and P- acquiring enzymes (including LAP and NP) were lowered by conventional ZnO-NPs irrigation, suggesting that microbial N- and P-limitation were the major constraints relative to C limitation, which is likely to be caused by SOC (or DOC) enhancement induced by ZnO-NPs (Figure 2I–K) [48]. All the above suggested that conventional ZnO-NPs irrigation ultimately induced increases in soil *Eh*, pH, and SOC concentration and changed the microbial activities, among which SOC might aggravate ZnO-NPs aggregation to limit nutrition absorption, thereby inducing its ecotoxicity. The increase in Zn^2+^ release was also a key reason.

Due to the nanobubble-containing water having a higher Do concentration, it can increase the soil oxygen concentration, thereby affecting the ecotoxicity of metal oxide nanoparticles (Figure 2A) [27,28]. Our results showed that soil *Eh* was further enhanced by nanobubble-treated ZnO-NPs irrigation, accompanied by an increase in water-soluble phenol concentration as compared to conventional ZnO-NPs irrigation (Figure 2E,G). It appeared to be attributed to the fact that nanobubble-containing water provides more oxygen, -OH and O_2_^−^ [36,49]. The CAT activity showed an increased trend in soil with nanobubble-treated ZnO-NPs irrigation as compared to conventional ZnO-NPs irrigation, indicating that oxygen was also increased due to H_2_O_2_ decomposition (Figure 2H). The higher level of soil oxygen resulting from nanobubble-containing water could facilitate the release of root exudates [50]. This was a possible reason why the DOC concentration was increased in soil with nanobubble-treated ZnO-NPs irrigation in our study (Figure 2F). Correspondingly, the SOC concentration was further enhanced in the presence of DOC input (Figure 2M) [43]. Another important aspect was that the ZnO aggregation was remitted by the increased SOC (including DOC), such as cysteine, thereby relieving the nutrient limitation [51,52]. Based on the above, we speculated that nanobubble-treated ZnO-NP irrigation may reduce the ecotoxicity of ZnO-NP by mitigating the ZnO-NP aggregates. The combined results, among which the nanobubble-treated ZnO-NPs irrigation caused the decreased Zn concentration in soil, the increased Zn concentration in roots, and the increased shoot biomass, suggested that the ecotoxicity of ZnO-NPs could mainly depend on the barrier formed by aggregates inducing the limited nutrient uptake of plants [46]. In relation to conventional ZnO-NPs irrigation, nanobubble-treated ZnO-NPs irrigation induced a shift of microbial nutrient limitation from microbial N- and P-limitation relative to C limitation to C- and P-limitation relative to N limitation, which was reflected in higher C-acquiring enzyme activity and lower N-acquiring enzyme activity. In conclusion, the nanobubbles treatment further raised the soil *Eh* and SOC concentrations induced by ZnO-NPs, among which the SOC may reduce ZnO-NP aggregates, thereby remitting its ecotoxicity.

### 3.3. Nanobubble Treatment Decreased Oxidative Damage, Promoted Energy Metabolism, and Stimulated Root Activity to Remit the Biotoxicity of ZnO-NPs

The pros and cons of Zn coexist for plants, among which over-accumulated Zn may cause a 50% decrease in biomass [53]. Roots, as the soil-plant interface, play a critical role in the absorption of nutrition and water for plant growth. Meanwhile, the nanoparticulate material was absorbed by plants via the root system [19]. Our results showed that conventional ZnO-NPs irrigation decreased root activity (Figure 3A). Wang et al. and Li et al. also demonstrated that the higher levels of ZnO-NPs caused a reduction in root activity in maize and tomato (*Solanum lycopersicon* L.) [20,54]. The potential reasons for decreased root activity described above were explored. We found that ROS concentrations, including H_2_O_2_ and O_2_^−^, were increased in the roots of wheat seedlings under conventional ZnO-NPs irrigation, which is one of the main potential mechanisms of ZnO-NPs-induced phytotoxicity due to the absorption of Zn^2+^ ions by roots (Figure 3B,C) [19]. The higher level of MDA, as an indicator of membrane lipid peroxidation, also clearly demonstrated that the roots were subjected to oxidative damage (Figure 3D) [55]. A possible explanation for the above phytotoxicity is the ineffective regulation of the antioxidant enzyme system [56]. As expected, the activities of two antioxidant enzymes, including APX and GST, were markedly lowered in roots with conventional ZnO-NPs irrigation (Figure 3E). Meanwhile, the energy metabolism in roots is also closely related to root activity and antioxidant enzyme system mobilization [57]. In our study, as compared to the water irrigation control, conventional ZnO-NPs irrigation decreased the glucose concentration and changed the key enzyme activities related to glycolysis and the pentose phosphate pathway, among which it is also noteworthy to point out that AGPase activity increased while PFK activity decreased (Figure 4A,B). AGPase is responsible for the conversion between glucose-1-phosphate and ADP-glucose, which is a limiting enzyme of starch biosynthesis [58]; PFK catalyzes an irreversible reaction in glycolysis, namely the conversion of fructose-6-phosphate to fructose-1,6-bisphosphate [59]. Hence, the above changes illustrated that conventional ZnO-NPs irrigation hindered glycolysis, thereby impairing energy production.

Surprisingly, as compared to conventional ZnO-NPs irrigation, the nanobubble-treated ZnO-NPs irrigation invoked varied effects on ROS, as manifested by increased O_2_^−^ concentration, decreased H_2_O_2_ concentration, and a final decreased MDA concentration, suggesting that nanobubble treatment remitted the oxidative stress caused by ZnO-NPs (Figure 3B–D) [60]. As such, the root activity appeared to be enhanced. For the relatively higher level of O_2_^−^, we speculated it may be caused by soil oxygen input and delayed initiation of the antioxidant enzyme system (Figure 3E). This suggested that another ROS scavenging pathway, such as antioxidants, played an important role in the remission of oxidative stress in roots with nanobubble-treated ZnO-NPs irrigation [61]. Meanwhile, in our study, the higher level of soil oxygen input may cause glucose accumulation and active energy metabolism in roots irrigated with the nanobubble-treated ZnO-NPs solution, as manifested by the increases in the activities of PFK and G6PDH, promoting the glycolysis and pentose phosphate pathways to produce more energy (Figure 4A,B).

### 3.4. Nanobubble Treatment Did Not Yet Effectively Regulate the Biotoxicity of ZnO-NPs to Leaves via Changing Redox State and Energy Metabolism

Unlike the case in roots, the conventional ZnO-NPs irrigation only caused a decrease in O_2_^−^ concentration; however, unfortunately, the MDA concentration was raised, indicating that leaves were subjected to oxidative damage (Figure 3C,D) [55]. The lower O_2_^−^ concentration was derived from the higher SOD activity caused by ZnO-NPs treatment (Figure 3E). The conventional ZnO-NPs irrigation not only decreased the glucose concentration in roots but also decreased the glucose concentration in leaves (Figure 4A). Meanwhile, it was perhaps surprising that most key enzyme activities related to glycolysis appeared to have a decreased trend, except for PFK in leaves with conventional ZnO-NPs irrigation (Figure 4B). In combination with the decreased tendency for the most enzyme activities and glucose concentration, our results showed that glycolysis was restrained in leaves, despite PFK activity increasing.

Similarly, judging from the observed changes in the present study, although the O_2_^−^ concentration was increased, the MDA concentration was not further increased in leaves with the nanobubble-treated ZnO-NPs irrigation as compared to that with the conventional ZnO-NPs irrigation (Figure 3C,D). It suggested that nanobubble treatment did not aggravate oxidative stress. We inferred that not only the catalysis of antioxidant enzymes (including DHAR, MDHAR, and GR) in plants but also the antioxidants accordingly produced by them could play a critical role (Figure 3E) [62]. Meanwhile, the glucose accumulation in leaves with the nanobubble-treated ZnO-NPs irrigation was increased as compared to that with the conventional ZnO-NPs irrigation (Figure 4A). Unlike the case under conventional ZnO-NPs irrigation, an equal number of glycolysis-related enzymes with increased activities and decreased activities appeared in leaves with nanobubble-treated ZnO-NPs irrigation, among which the activities of two key enzymes, i.e., PFK and Ald (another enzyme controlling an irreversible step in glycolysis), were suppressed (Figure 4B). Hence, these results suggested that nanobubbles could not have a similar promoting effect on the glycolysis and pentose phosphate pathways in leaves as they had in roots.

### 3.5. Nanobubble Treatment Conferred the Advantage of a Rapid Response to the Biotoxicity of ZnO-NPs via the Trade-off of Leaf Functional Traits, Especially Nutrient Contents

The leaf trait data collected from 2548 species and 175 sites thus far provides strong evidence for the existence of functional integration for certain foliar traits, mainly including *A*_area_, *R*_dark,area_, *N*_area_, *P*_area_, LMA, and leaf life span (LL) [63,64]. Although vascular plant species have different growth patterns and environmental affinity, the trade-off relationships between the costs of leaf structure and the time of resource return were quantificationally determined via the above foliar traits, forming a continuously variable species spectrum (referred to as the leaf economics spectrum) [65]. On one end of this species spectrum are slow investment-return species with long-lived leaves that use resources conservatively and have low photosynthetic capacity; at the opposite end of this species spectrum are quick investment-return species with short-lived leaves that have high photosynthetic capacity and rates of resource use [64]. The previous studies showed that environmental changes may prompt the occurrence of trade-off strategies in plants, further forming the leaf economic spectrum [63,66]. In our study, conventional ZnO-NP irrigation observably increased the *A*_area_, but decreased the LMA, suggesting the wheat plants were more prone to becoming quick investment-return species under ZnO-NP treatment (Figure 5A,B). We inferred that leaf N was theoretically enhanced by conventional ZnO-NP irrigation because it was essential to synthesize the proteins of photosynthetic machinery, thereby facilitating the photosynthetic rate [67]. However, we further found that the conventional ZnO-NP irrigation observably decreased the N_area_ but increased the PNUE, *PN*_et_, *PN*_cb_, and *J*_max_ (Figure 5D and Figure 6J,K). The above results showed that ZnO-NPs increased the *A*_area_ via enhancing *J*_max_, with the increases in PNUE and *PN*_et_ determining the increased *J*_max_ [68,69,70]. Interestingly, the microbial N-limitation also occurs in soil irrigated with the conventional ZnO-NPs solution (Figure 2J,K). The limitations of microbial N and leaf N provided possible lines of evidence that the ZnO-NPs aggregate-induced nutrient uptake limitation was a major reason for the ZnO-NPs ecotoxicity. The *V*_cmax_ was not markedly affected by conventional ZnO-NPs irrigation, suggesting CO_2_ concentration was not a limiting factor. Hence, the decreased *C*_i_ did not decrease *A*_area_ (or *A*_n_) (Figure 6E) [71]. From the LMA perspective, the decreased LMA in leaves caused by the conventional ZnO-NPs irrigation was ascribed to the decreases in leaf N and *PN*_non-psn_ (Figure 5D and Figure 6K). In addition, the leaves with higher LMA have a greater thickness to increase the CO_2_ diffusion distance, and the *g*_s_ were slightly decreased, which could be responsible for the decreased *C*_i_ (Figure 5F and Figure 6C,E) [64,72]. The intensively studied roles of photosystem II were conducted, and it was found that PIabs and DI_O_/RC were decreased, suggesting the ZnO-NPs restrained the excess energy dissipation of the reaction center, thereby inducing damage to photosystem II (Figure 6H) [73]. A similar result was found in the study of Rai-Kalal and Jajoo [74].

Of note, the nanobubble-treated ZnO-NPs irrigation induced a slight increase in *A*_area_ and prominent increases in *N*_area_ and *P*_area_ as compared to conventional ZnO-NP irrigation (Figure 5A,D,E). These results illustrated that wheat plants with nanobubble-treated ZnO-NP irrigation further became established as quick-investment-return species. Meanwhile, we amazedly found that the microbial N-limitation was relieved, which is similar to leaf *N*_area_ (Figure 2J,K). However, LMA was increased by the nanobubble-treated ZnO-NPs irrigation, implying that the higher levels of *N*_area_ and *P*_area_ not only increased the processes of quick return on investments (e.g., photosynthesis and metabolic consumption of photosynthate), but also increased the processes of slow return on investments (e.g., leaf structure construction) (Figure 5A) [64]. It was also well reflected in the increase of *PN*_non-psn_ (Figure 6K). In addition to *PN*_non-psn_, the PNUE, *PN*_et_, and *PN*_cb_ were affected by the nanobubble-treated ZnO-NPs irrigation, which decreased in relation to the conventional ZnO-NPs irrigation, thereby causing the decline of *J*_max_ (Figure 6G,J,K). This is not surprising because, along with significantly increased *N*_area_, *A*_area_ only exhibited a slight increase in wheat seedlings with the nanobubble-treated ZnO-NPs. Meanwhile, we found that a significant change in total chlorophyll concentration did not affect light-harvesting (Figure 6A). The increased *PI*_abs_ indicated the nanobubble-treated ZnO-NPs irrigation indeed remitted the ZnO stress (Figure 6I).

### 3.6. Nanobubble Treatment Confers the Advantage of a Rapid Response to the Biotoxicity of ZnO-NPs via Water Potential

Water status has an intimate interaction with photosynthetic performance [74]. Water use efficiency, as a parameter representing the relationship between matter accumulation and water consumption, is dependent on many physiological processes, among which photosynthesis is the most direct impact factor [75,76]. Due to the slight increases in *g*_s_ and *T*_r_, the higher *A*_n_ induced the increases in both WUE_i_ and WUE_inst_, suggesting that wheat seedlings enhanced ZnO-NPs resistance per se via increasing water availability in leaves (Figure 6B–D and Figure 7A). Water potential represents the water transport capacity of plants. The water flow is taken up by roots, then enters the plant's vascular system and transpires into the atmosphere through stomata [77]. This process depends on a gradient in water potential, which is dominated by transpiration. The prevailing consensus is that a more negative leaf water potential (Ψ_leaf_) is indispensable for maintaining a higher transpiration rate [78,79]. Precisely, our results also showed that Ψ_leaf_ was decreased under conventional ZnO-NPs irrigation, along with a slight increase in *T*_r_ (Figure 6D and Figure 7B). Overall, wheat seedlings resisted the biotoxicity of ZnO-NPs by enhancing water use efficiency (WUE) and lowering the Ψ_leaf_ to maintain a high level of water evaporation. Under the nanobubble-treated ZnO-NPs irrigation, the Ψ_leaf_ was increased because the transpiration rate was slightly decreased as compared to the conventional ZnO-NPs irrigation (Figure 6D and Figure 7B). This result illustrated that the nanobubble-treated ZnO-NPs irrigation decreased water loss and accordingly regulated the Ψ_leaf_ to reduce the biotoxicity of ZnO-NPs.

## 4. Materials and Methods

### 4.1. ZnO Nanoparticles and Plant Materials

The ZnO-NPs (average size: 20 nm, purity: 99.9%) were purchased by XFNANO Materials Tech Co., Ltd. (Nanjing, China). The spring wheat cv. Novosibirskaya 67 sown in pots was cultivated in a climate chamber under the following conditions: 24/19 °C day/night temperature, 65–70% relative humidity, and >500 μmol m^−2^ s^−1^ photosynthetic active radiation (PAR).

### 4.2. Experimental Design

Wheat seeds were surface sterilized with 2% HClO for 30 min, followed by triple rinses with ultrapure water. Six sterilized seeds were sown in each plastic pot (10 cm in height and 15 cm in diameter) and filled with 1.3 kg of pre-prepared black soil. The pots were divided into two groups, with one group irrigated with water and the other group irrigated with nanobubble-containing water. Nanobubble water was generated using a nanobubble generator (ZJ Environment Protection, Shanghai, China) in combination with mechanical shear and depressurization [36]. The generator, with a power of 300 W and a flow rate of 0.2 m^3^ h^−1^, produced nanobubbles with a mean size of approximately 74 nm according to the manufacturer’s specifications. Nanobubble-containing water was immediately utilized after preparation.

ZnO-NPs treatment was initiated when wheat seedlings reached the tillering stage. In each group, half of the plants were irrigated with 500 mg L^−1^ ZnO-NPs solution (ZnO nanoparticles mixed within water or nanobubble-containing water, and the concentration of ZnO-NPs was determined based on Guo et al. [21]), while the remaining half continued to receive normal water or nanobubble-containing water irrigation conventionally. In summary, there were four irrigation treatments at the tillering stage in this study: ZnO_0_W, treatment with water; ZnO_0_nW, treatment with nanobubble-containing water; ZnO_500_W, treatment with 500 mg L^−1^ ZnO-NPs-containing water; and ZnO_500_nW, treatment with 500 mg L^−1^ ZnO-NPs- and nanobubble-containing water. Seven days post-treatment, both the soil and wheat leaves were measured and collected for further analysis (Figure 8). 

### 4.3. Determination of Do

Dissolved oxygen (Do) levels were determined using a multi-parameter analyzer equipped with a Do electrode (Shanghai INESA SCIENTIFIC Instrument, Shanghai, China). The electrode was immersed in each treatment solution in a continuous measurement mode. Each treatment was replicated five times.

### 4.4. Soil Properties

In situ, soil *Eh* measurement was conducted using an ORF analyzer (Shanghai INESA SCIENTIFIC Instrument, Shanghai, China). Soil pH was measured by a multi-parameter analyzer with a pH electrode (Shanghai INESA SCIENTIFIC, Shanghai, China). A soil sample of 10 g was transferred into beakers with 25 mL of deionized water, stirred for 1 min, and left to stand for 30 min to reduce the CO_2_ content. The soil pH was measured. This experiment was replicated three times.

SOC concentration was measured using the SOC kit (Suzhou Comin Biotechnology, Suzhou, China) following the manufacturer’s instructions. The DOC concentration was quantified through a spectrophotometric assay. Briefly, 10 g of fresh soil was mixed well with deionized water at a ratio of 1:5 (*w*/*v*), and centrifuged at 15,000× *g* for 20 min. The supernatant was further filtered through a 0.45 μm filter. 2 mL of the leachate was mixed with a buffer containing 3 mL deionized water, 2.5 mL Mn(III)-pyrophosphoric acid (10 mM), and 2.5 mL H_2_SO_4_, and let stand for 1 h. The absorbance was recorded at 500 nm (Liu et al., 2022) [41]. This experiment was replicated three times. 

The water-soluble phenol concentration was determined using the Folin-Ciocalteau method [80]. A dry soil sample of 5 g was mixed with 25 mL of deionized water in a 50 mL conical flask. After 2 h of shaking and 24 h of settling, the mixture was shaken for another 30 min and then centrifuged for 15 min at 6000× *g*. 2 mL of supernatant was mixed with 2 mL of Folin and Ciocalteu’s phenol reagent and 6 mL of Na_2_CO_3_. After 15 min of incubation at 37 °C, the mixture was measured at 680 nm. This experiment was replicated three times.

Following the method of Li et al. [43], soil Fe (II) was extracted from dry soil using 0.5 M HCl. After 2 h of shaking at 25 °C, the solution was centrifuged for 10 min at 240× *g*. 2 mL of extract was mixed with 0.2 mL of deionized water, which was then mixed with 0.2 mL of phenanthroline (0.15%), 0.2 mL of HAc-NaAc, and 0.4 mL of deionized water. Total iron concentration was measured using the same method, except that 0.2 mL of hydroxylamine hydrochloride (10%) was added to the extraction buffer. The ferric iron (Fe(III)) concentration was calculated based on the concentrations of total iron and Fe(II) [80,81]. This experiment was replicated three times.

### 4.5. Measurement of Soil Enzyme Activities and Enzymatic Stoichiometry

The activities of five soil enzymes were determined spectrophotometrically using an enzymatic activity analysis kit (Beijing Boxbio Science and Technology, Beijing, China), including one C-acquisition enzyme (BG), two N-acquisition enzymes (β-1,4-*N*-acetylglucosaminidase, NAG; LAP), one P-acquisition enzyme (NP), and CAT. Finally, the enzyme activities were expressed as nmol g^−1^ h^−1^. Enzymatic stoichiometry was employed to reflect microbial nutrient status [82,83]. Here, the stoichiometric ratios of extracellular enzymes were calculated by Ln (BG): Ln (LAP + NAG), Ln (BG): Ln (AP), and Ln (LAP + NAG): Ln (AP) [84]. Each treatment had three replicates.

### 4.6. Measurement of Zn Concentrations in Soil and Roots

The soil sample and root sample were transferred into a pressure tank containing HNO_3_. Incubation was then carried out in stages. The extracts were then analyzed using inductively coupled plasma-mass spectrometry (ICP-MS, Thermo Scientific, Waltham, MA, USA). Each treatment had four replicates.

### 4.7. Measurement of Plant Phenotype

The shoot's fresh weight and plant height were recorded. Leaf thickness was measured using MultispeQ (Photosynq, East Lansing, MI, USA), and leaf area was determined with a root scanner (Epson, Shanghai, China). Subsequently, all leaves and the left parts of shoots were separately dried at 70 °C for 72 h before recording their dry weights. LMA was calculated based on leaf area and the corresponding leaf dry weight. The shoot dry weight was the sum of the leaf dry weight and the weight of the left parts of the shoot. 

### 4.8. Measurements of Gas Exchange Parameters and the Response Curve

After light adaptation, leaves were clamped with the LED leaf chamber of the portable LI-6400 photosynthesis system (Li-Cor Biosciences, Lincoln, NB, USA). The CO_2_ concentration was set at 400 μmol mol^−1^ using a CO_2_ mixture, PAR was 1200 μmol m^−2^ s^−1^, the ambient temperature was 25 °C, and the flow rate was maintained at 500 μmol s^−1^. The measurement was conducted for 20 min to ensure stabilization. Gas exchange parameters were recorded when the steady state was achieved. Subsequently, a light response curve was generated by maintaining light intensity in the leaf chamber across a series of levels: 2000, 1500, 1200, 1000, 750, 500, 250, 150, 100, 60, 20, 0 μmol m^−2^ s^−1^. The leaves were equilibrated for 2–3 min at each light intensity. The CO_2_ response curve was obtained after 40 min of photoinduction at a PAR of 1800 μmol m^−2^ s^−1^. CO_2_ concentrations surrounding the leaf were set at 400, 300, 200, 150, 100, 50, 400, 600, 800, 1000, 1500, and 2000 μmol mol^−1^. The leaves were equilibrated for 2–3 min at each CO_2_ concentration. 

### 4.9. Measurement of WUE and Leaf Water Potential

According to the gas exchange parameters, the WUE was calculated using the following Equations (1) and (2): [85,86]
WUE_i_ = *A*_n_/*g*_s_(1)
WUE_inst_ = *A*_n_/*T*_r_(2)
WUE_i_ and WUE_inst_, intrinsic water-use efficiency and instantaneous water-use efficiency; *A*_n_, net photosynthetic rate; *g*_s_, stomatal conductance; *T*_r_, transpiration rate. Each treatment had three replicates.

The leaf water potential was determined with the chilled-mirror dewpoint method using a dewpoint potentiometer (Decagon Devices, Pullman, WA, USA) [87,88]. Each treatment had three replicates.

### 4.10. Measurement of Chlorophyll a Fluorescence and Photosynthetic Pigment Concentrations

Chlorophyll *a* fluorescence transient (namely the OJIP curve) was performed using a portable fluorometer (Photon System Instruments, Drasov, Czech Republic). The quantitative detection of the photosynthetic pigments, including chlorophyll and carotenoid, was processed using the modified acetone extraction method, according to Lichtenthaler [89]. Each treatment had three replicates.

### 4.11. Measurement of Root Activity, the Concentrations of ROS and MDA, and Antioxidant Enzyme Activities

The roots were gently washed using distilled water, and then the root activity was measured using the triphenyltetrazole chloride (TTC) method. [59] The concentrations of H_2_O_2_, O_2_^−^, and MDA in leaves and roots were determined according to the instructions of the assay kits (Solarbio Life Sciences, Beijing, China). The antioxidant enzyme activities in roots and leaves, including SOD, peroxidase (POX), cell wall peroxidase (cwPOX), APX, CAT, DHAR, MDHAR, GR, and GST, were estimated by kinetic assays using an Epoch Take3 spectrophotometer (BioTek Instruments, Inc., Winooski, VT, USA) with a 96-well microtiter format [90]. Each treatment had three replicates.

### 4.12. Measurement of Glucose Concentration and Carbohydrate Metabolism Enzyme Activities

To explore how different forms of ZnO-NP irrigation affect the energy metabolism, the glycolysis and pentose phosphate pathways were tested. Glucose concentrations in leaves and roots were determined using a plant glucose assay kit (Solarbio Life Sciences, Beijing, China). Thirteen kinds of key carbohydrate metabolism enzyme activities, including HXK, Ald, FK, G6PDH, PGM, UGPase, AGPase, PFK, PGI, sucrose synthase (SuSy), vacInv, cwInv, and cytInv, were measured by kinetic enzyme assays. The dynamics of absorbance at a given wavelength for 40 min were monitored using an Epoch Take3 spectrophotometer (BioTek Instruments, Inc., Winooski, VT, USA) with a 96-well microtiter format. The specific enzyme activity was calculated within the linear range of substrate conversion and expressed in nkat mg^−1^ FW [91]. Each treatment had three replicates.

### 4.13. Measurement of the Concentrations of Leaf N and P, PNUE, and N Allocation within the Photosynthetic Apparatus

The N and P concentrations of the dry sample were measured using the micro-Kjeldahl method and the molybdenum blue method, respectively [92,93]. 

According to Wang et al., the PNUE was given by: [94]
PNUE = *A*_nmax_/*N*_area_(3)
*A*_nmax_, the maximum net photosynthetic rate; *N*_area_, total foliar N.

N allocation within the photosynthetic apparatus was calculated according to the model from Niinemets et al. [95].
*PN*_cb_ = *V*_cmax_/(6.25 × *V*_cr_ × *N*_area_)(4)
*PN*_cb_, the fraction of nitrogen allocated to the carboxylation system; *V*_cmax_, maximum ribulose-1,5-bisphosphate carboxylase/oxygenase (Rubisco) carboxylation rate; 6.25, 6.25 g Rubisco (g nitrogen in Rubisco)^−1^ converts nitrogen content to protein content; *V*_cr_, the maximum rate of Ribulose-1,5-bisphosphate (RUBP) carboxylation per unit Rubisco protein (20.8 μmol CO_2_ (g Rubisco)^−1^ s^−1^); *N*_area_, total foliar N.
*PN*_et_ = *J*_max_/(8.06 × *J*_mc_ × *N*_area_)(5)
*PN*_et_, the fraction of nitrogen allocated to electron transport components; *J*_max_, maximum electron transport rate; 8.06, the investment in bioenergetics is at least 8.06 μmol cyt f (g N in bioenergetics)^−1^; *J*_mc_, the potential rate of photosynthetic electron transport per unit cytochrome f (155.6 μmol electrons (μmol cyt f)^−1^ s^−1^); *N*_area_, total foliar N.
*PN*_lc_ = *C*_C_/(*N*_area_ × *C*_B_)(6)
*PN*_lc_, the fraction of nitrogen allocated to the light capture system; *C*c, Total chlorophyll concentration; *N*_area_, total foliar N; *C*_B_, chlorophyll-binding (2.15 mmol (g N)^−1^).
*PN*_non-psn_ = 1 − (*PN*_lc_ + *PN*_et_ + *PN*_cb_)(7)
*PN*_non-psn_ is the fraction of nitrogen allocated to non-photosynthetic nitrogen.

### 4.14. Statistical Analyses

All the data underwent testing for homogeneity of variance and normality of distribution. A two-way ANOVA was conducted, followed by the Duncan test at *p* ≤ 0.05 to compare differences between individual treatments in SPSS 22.0 (SPSS Inc., Chicago, IL, USA). The light response curves and A/Ci were analyzed in R (v.4.1.0; R Core Team, 2020).

## 5. Conclusions

In summary, a high level of ZnO-NPs exerted ecotoxicity on wheat plants, which was evidenced by reduced shoot fresh and dry weights. On one hand, ZnO-NPs induced increases in soil *E*h and SOC through rhizosphere effects associated with ROS, leading to Fenton reaction-induced SOC mineralization (Figure 9). This raises the possibility that the formation of ZnO aggregates may trigger limitations in nutrient absorption and microbial N and P. On the other hand, the diminished release of Zn^2+^ from ZnO-NPs induced ROS, including H_2_O_2_ and O_2_^−^, to decrease root activity and inhibit glycolysis in roots and leaves. The limited nutrient absorption led to a decline in leaf N. As a result, wheat plants increased the *A*_area_ (or *A*_n_) through elevated *J*_max_ due to rises in PNUE and *PN*_et_. The nanobubble irrigation contributed to mitigating the ZnO-NPs ecotoxicity to crops. With the increase in SOC concentration, nanobubble irrigation inhibited ZnO aggregation, potentially removing the obstacle to nutrient absorption. Moreover, nanobubble irrigation enhanced the root activity and glycolysis efficiency, as well as increased the leaf *N*_area_ and *P*_area_, thereby boosting the construction investment. This study proposes that nanobubble treatment presents a promising technique for mitigating the ZnO-NPs ecotoxicity.

## Figures and Tables

**Figure 1 ijms-25-01896-f001:**
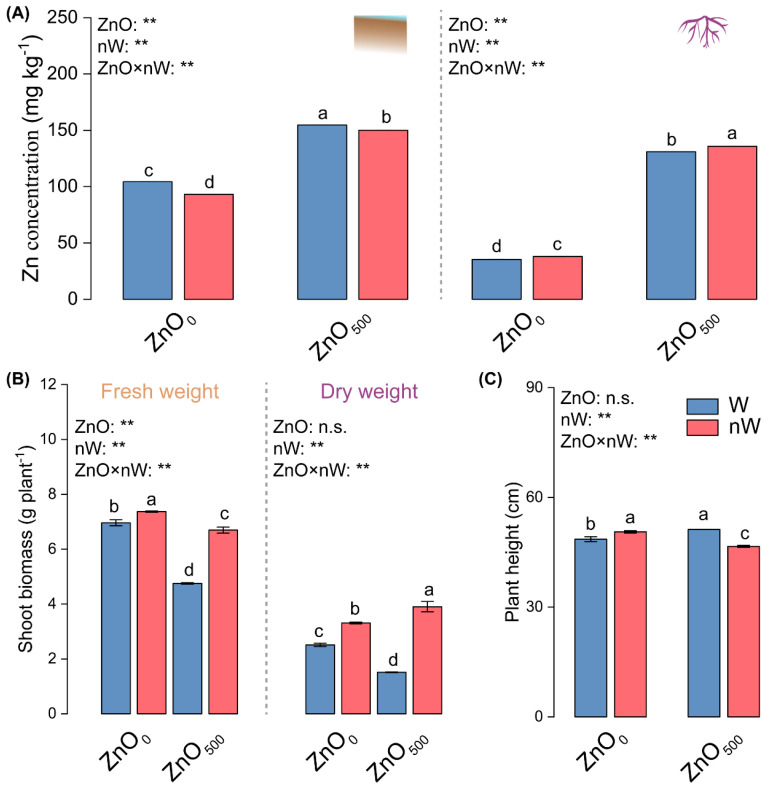
Zn concentration and plant performance in wheat under different irrigation treatments. (**A**) Zn concentration in soil and roots; (**B**) Shoot fresh weight and dry weight; (**C**) Plant height. Vertical bars indicate mean ± SE (*n* = 3). ZnO_500_ and ZnO_0_ indicate irrigation with/without 500 mg L^−1^ ZnO solution, while W and nW indicate normal water and nanobubble-containing water. n.s., no significant difference; **, *p* < 0.01. Different small letters indicate significant differences at *p* < 0.05 level.

**Figure 2 ijms-25-01896-f002:**
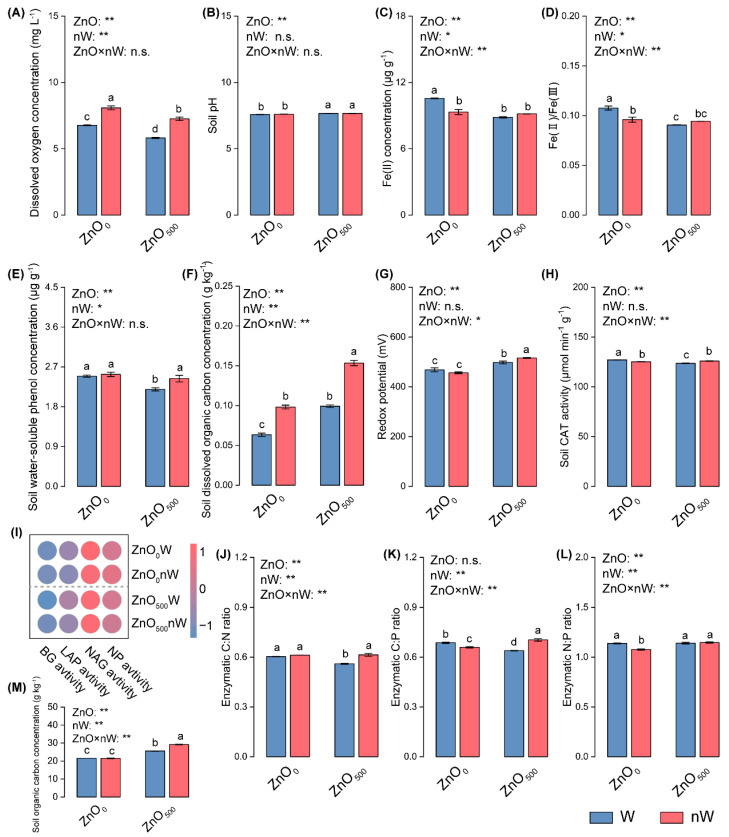
Soil physicochemical characteristics and activities of C-, N- and P-degrading enzymes and their stoichiometry under different irrigation treatments. (**A**) Dissolved oxygen concentration of different irrigation water; (**B**) Soil pH; (**C**) Soil ferrous (Fe(II)) concentration; (**D**) The ratio of ferrous to ferric iron (Fe(II)/Fe(III)); (**E**) Soil water-soluble phenol concentration; (**F**) Soil dissolved organic carbon concentration; (**G**) Redox potential; (**H**) Soil catalase (CAT) activity; (**I**) Heat map of extracellular enzyme activities in soil. The difference in activity for a given enzyme between the control (ZnO_0_W) and treatment is normalized and converted to a color scale. Increase and decrease in activity are indicated in red and blue, respectively. (**J**–**L**) The C:N:P stoichiometry of extracellular enzymes; (**M**) Soil organic carbon concentration. Vertical bars indicate mean ± SE (Dissolved oxygen concentration, *n* = 5; other parameters, *n* = 3). Abbreviations for treatments are explained in Figure 1. n.s., no significant difference; **, *p* < 0.01; *, *p* < 0.05. Different small letters indicate significant differences at *p* < 0.05 level. BG, β-d-glucosidase; LAP, L-leucine aminopeptidase; NAG, β-1,4-*N*-acetylglucosaminidase; NP, neutral phosphatase.

**Figure 3 ijms-25-01896-f003:**
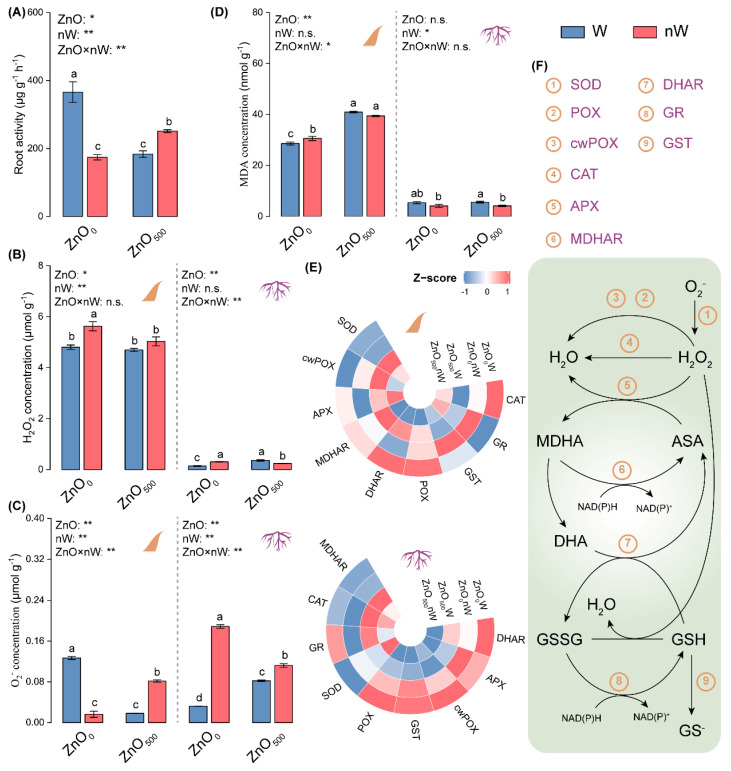
Root activity, reactive oxygen species (ROS) metabolism, and malondialdehyde (MDA) concentration in wheat under different irrigation treatments. (**A**) Root activity; (**B**) H_2_O_2_ concentration in leaves and roots; (**C**) O_2_^−^ concentration in leaves and roots; (**D**) MDA concentration in leaves and roots; (**E**) Heat map of activities of antioxidant enzymes in leaves and roots. The difference in activity for a given enzyme between the control (ZnO_0_W) and treatment is normalized and converted to a color scale. Increase and decrease in activity are indicated in red and blue, respectively. (**F**) Graphics showing the antioxidant enzyme system. Vertical bars indicate mean ± SE (*n* = 3). Abbreviations of treatments are explained in Figure 1. n.s., no significant difference; **, *p* < 0.01; *, *p* < 0.05. Different small letters indicate significant differences at *p* < 0.05 level. APX, ascorbate peroxidase; ASA, ascorbic acid; CAT, catalase; cwPOX, cell wall peroxidase; DHA, dehydroascorbic acid; DHAR, dehydroascorbate reductase; GR, glutathione reductase; GSH, reduced glutathione; GSSG, oxidized glutathione; GST, glutathione S-transferase; MDHA, monodehydroascorbate; MDHAR, monodehydroascorbate reductase; POX, peroxidase; SOD, superoxide dismutase.

**Figure 4 ijms-25-01896-f004:**
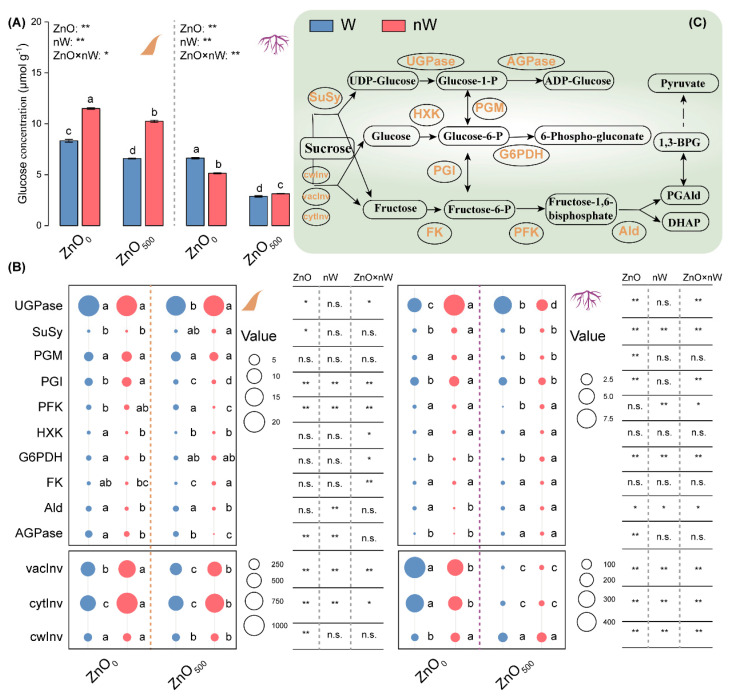
Glycolysis and pentose-phosphate pathways in wheat under different irrigation treatments. (**A**) Glucose concentration in leaves and roots; (**B**) Activities of key enzymes related to glycolysis and pentose phosphate pathway in leaves and roots; (**C**) Graphics showing the glycolysis and pentose phosphate pathways. Vertical bars indicate mean ± SE (*n* = 3). Abbreviations for treatments are explained in Figure 1. n.s., no significant difference; **, *p* < 0.01; *, *p* < 0.05. Different small letters indicate significant differences at *p* < 0.05 level. AGPase, ADP-glucose pyrophosphorylase; Ald, Aldolase; DHAP, dihydroxyacetone phosphate; Fructonse-6-P, fructonse-6-phosphate; Glucose-1-P, glucose-1-phosphate; Glucose-6-P, glucose-6-phosphate; G6PDH, glucose-6-phosphate dehydrogenase; PFK, phosphofructokinase; PGAld, glyceraldehyde-3-phosphate; PGI, phosphoglucoisomerase; PGM, phosphoglucomutase; UGPase, UDP-glucose pyrophorylase; 1,3-BPG, 1,3-bisphosphoglycerate.

**Figure 5 ijms-25-01896-f005:**
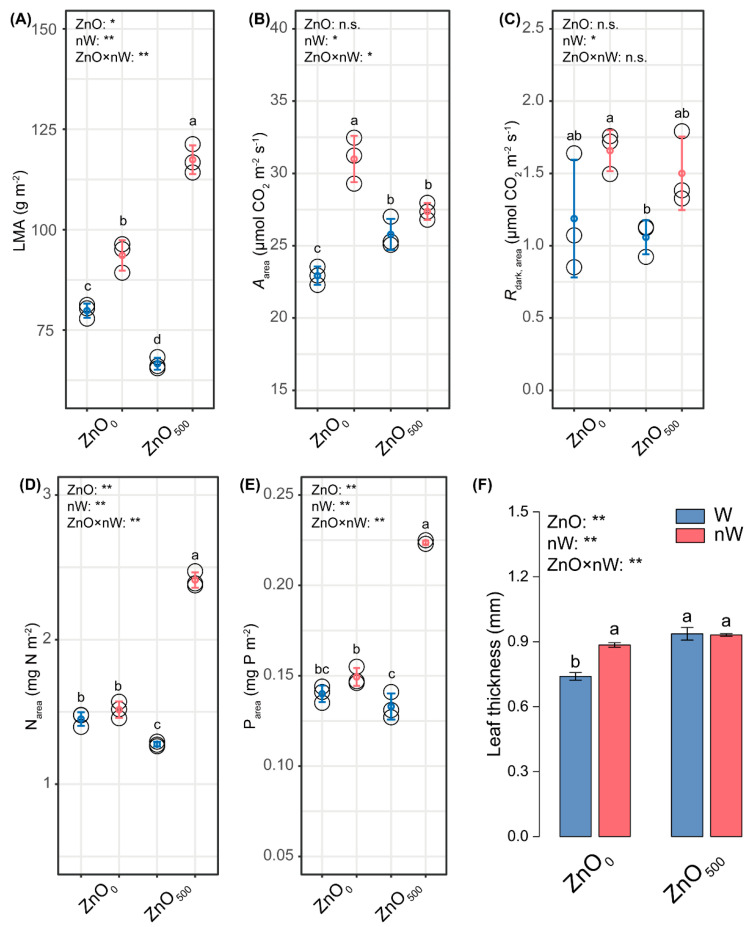
Leaf economics spectrum in wheat under different irrigation treatments. (**A**) Leaf dry mass per area (LMA); (**B**) Photosynthetic capacity (*A*_area_); (**C**) Dark respiration rate (*R*_dark_, area); (**D**) Total foliar nitrogen (*N*_area_); (**E**) Total foliar phosphorus (*P*_area_); (**F**) Leaf thickness. Vertical bars indicate mean ± SE (*n* = 3). Abbreviations of treatments are explained in Figure 1. n.s., no significant difference; **, *p* < 0.01; *, *p* < 0.05. Different small letters indicate significant differences at *p* < 0.05 level.

**Figure 6 ijms-25-01896-f006:**
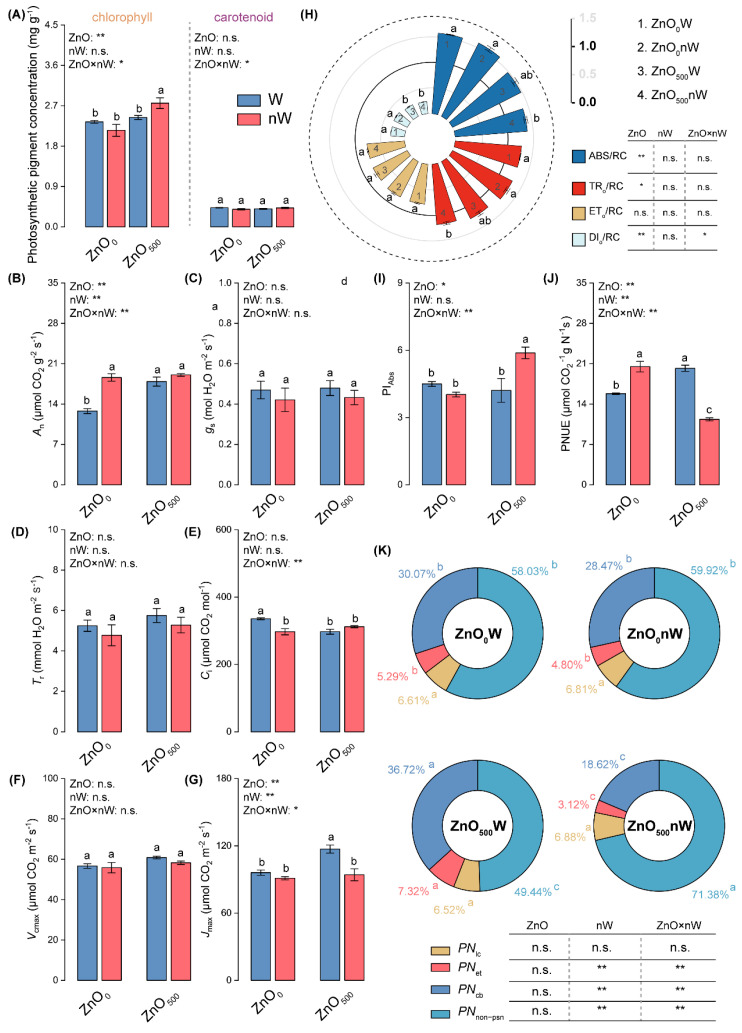
Photosynthetic performance of wheat under different irrigation treatments. (**A**) Photosynthetic pigment concentration; (**B**) Net photosynthetic rate (*A*_n_); (**C**) Stomatal conductance (*g*_s_); (**D**) Transpiration rate (*T*_r_); (**E**) Intercellular CO_2_ concentration (*C*_i_); (**F**) Maximum ribulose-1,5-bisphosphate carboxylase/oxygenase carboxylation rate (*V*_cmax_); (**G**) Maximum electron transport rate (*J*_max_); (**H**) Specific energy fluxes (per QA-reducing photosystem II reaction center); (**I**) Performance index (*PI*_abs_); (**J**) Photosynthetic nitrogen use efficiency (PNUE); (**K**) Nitrogen allocation within the photosynthetic apparatus. Vertical bars indicate mean ± SE (*n* = 3). Abbreviations of treatments are explained in Figure 1. n.s., no significant difference; **, *p* < 0.01; *, *p* < 0.05. Different small letters indicate significant differences at *p* < 0.05 level. ABS/RC, absorption flux per reaction center (RC); DI_O_/RC, dissipated energy flux per RC; ET_o_/RC, electron transport flux per RC; *PN*_cb_, the fraction of nitrogen allocated to carboxylation system; *PN*_et_, the fraction of nitrogen allocated to electron transport components; *PN*_lc_, the fraction of nitrogen allocated to light capture system; *PN*_non-psn_, the fraction of nitrogen allocated to non-photosynthetic nitrogen; TR_o_/RC, trapped energy flux per RC.

**Figure 7 ijms-25-01896-f007:**
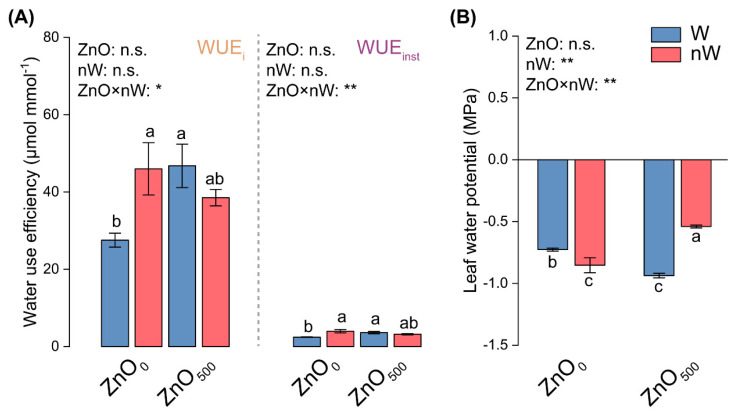
Water use efficiency and water potential in wheat under different irrigation treatments. (**A**) Intrinsic water-use efficiency (WUE_i_) and instantaneous water-use efficiency (WUE_inst_). (**B**) Leaf water potential (Ψ_leaf_). Vertical bars indicate mean ± SE (*n* = 3). Abbreviations of treatments are explained in Figure 1. ns, no significant difference; **, *p* < 0.01; *, *p* < 0.05. Different small letters indicate significant differences at *p* < 0.05 level.

**Figure 8 ijms-25-01896-f008:**
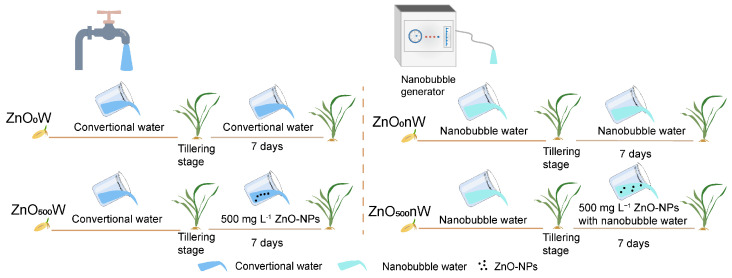
The diagram of the experimental arrangements.

**Figure 9 ijms-25-01896-f009:**
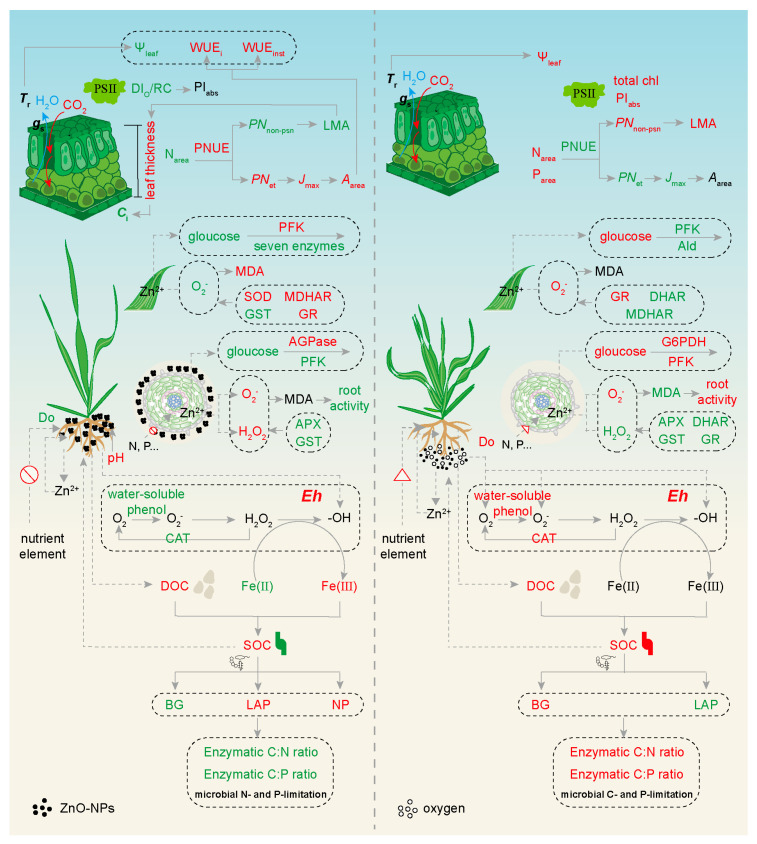
Comprehensive description of the underlying mechanisms of Nanobubble watering, mitigating the ecotoxicity of ZnO-NPs. The dashed line represents the possible regulatory pathways. Increased enzyme activities and metabolite concentrations are indicated in red, while decreased ones are indicated in green. *A*_area_, photosynthetic capacity; AGPase, ADP-glucose pyrophosphorylase; Ald, aldolase; APX, ascorbate peroxidase; BG, β-D-glucosidase; CAT, soil catalase; *C*_i_, intercellular CO_2_ concentration; DHAR, dehydroascorbate reductase; Do, dissolved oxygen; DOC, soil dissolved organic carbon; DI_O_/RC, dissipated energy flux per reaction center; *Eh*, redox potential; Fe(II), soil ferrous; Fe(III), ferric iron; GR, glutathione reductase; *g*_s_, stomatal conductance; GST, glutathione S-transferase; G6PDH, glucose-6-phosphate dehydrogenase; *J*_max_, maximum electron transport rate; LAP, L-leucine aminopeptidase; LMA, leaf dry mass per area; MDA, malondialdehyde; MDHAR, monodehydroascorbate reductase; N_area_, total foliar N; NP, neutral phosphatase; P_area_, total foliar P; PFK, phosphofructokinase; PI_abs_, performance index; *PN*_et_, the fraction of nitrogen allocated to electron transport components; *PN*_non-psn_, the fraction of nitrogen allocated to non-photosynthetic nitrogen; PNUE, photosynthetic nitrogen use efficiency; PSII, photosystem II; SOC, soil organic carbon; SOD, superoxide dismutase; Total chl, total chlorophyll; *T*_r_, transpiration rate; WUE_i_, intrinsic water-use efficiency; WUE_inst_, instantaneous water-use efficiency; Ψ_leaf_, leaf water potential.

## Data Availability

The data presented in this study are available on request from the corresponding author.

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
