# Peer review of "An Innovative Approach to Alleviate Zinc Oxide Nanoparticle Stress on Wheat through Nanobubble Irrigation"

_ijms, 2024, doi:10.3390/ijms25031896_

Round 1

Reviewer 1 Report

Comments and Suggestions for Authors

The study of Feng Zhang and co-authors entitled “An Innovative Approach for Rational Utilization of Zinc Oxide Nanoparticle to Enhance Stress Tolerance of Wheat Through Nanobubble Irrigation” is relevant and provides valuable insights into the influence of nanobubble watering on soil quality and crop production and offers an innovative approach for agricultural irrigation that enhances the effectiveness and efficiency of water application as well as presents a promising technique for mitigating the ZnO-NPs ecotoxicity.

The study is comprehensive, well-planned, and covers a wide range of indicators including Zn concentration and plant performance, water quality property, soil physicochemical characteristics and enzymatic properties, plant stress damage and antioxidant enzyme activities, leaf functional traits, photosynthetic parameters, and etc. The obtained results well-presented and discussed. The paper will be of great interest to a wide range of readers since it has a large scientific and practical significance.

In my opinion, the paper must be accepted for publication after double-checking the text (for example, the titles of subsections 2.3 and 2.4 are the same, it must be checked; lines 91, 93, 158 – also should be checked).

Author Response

Dear editor,

We are grateful for the valuable and constructive comments posted by you and the reviewers on the manuscript (Manuscript ID: ijms-2851286). We have considered all of the comments carefully and made improvement accordingly. We hope the revised manuscript will be acceptable for publication on International Journal of Molecular Science.

Best regards.

Xiangnan Li

Response to reviewer comments:

Reviewers 1:

Response to overall comments: Thanks for the comments, the manuscript has been improved again.

Comment 1: In my opinion, the paper must be accepted for publication after double-checking the text (for example, the titles of subsections 2.3 and 2.4 are the same, it must be checked; lines 91, 93, 158 – also should be checked).

Response 1: Thank you for the kind comments. The text has been re-checked carefully again. The titles of subsection 2.3 and 2.4 and format errors have been remodified in new version (for example, L91, L93, L105, L147, L153, L193). 

Reviewer 2 Report

Comments and Suggestions for Authors

The submitted manuscript to IJMS entitled “An Innovative Approach for Rational Utilization of Zinc Oxide Nanoparticle to Enhance Stress Tolerance of Wheat Through Nanobubble Irrigation” is of great potential to be published. But, before publication, following are the comments that need to be addressed:

Why did the authors use *rational* in the title?

It would be nice to show the experimental system visually so that the readers get interested.

The title should be revised completely since the authors provide nanobubble as a solution against Zinc Oxide Nanoparticle toxicity.

How did the authors choose 500 mg 569 L−1 ZnO-NPs? Were there any preliminary experiments?   

Line 670: It is not enough to provide the reference. It is mandatory to explain the applied methods for all the parameters.

Line 704: How can the authors have concluded about the inhibition of glycolysis? Did they measure all the relevant metabolites related to glycolysis and adjacent processes?

Authors did a very good job in presenting their study however, there are some major concerns in the title and explanation of the experimental design. Therefore, I strongly suggest the authors improve them.  

Comments on the Quality of English Language

Moderate editing of English language required.

Author Response

Dear editor,

We are grateful for the valuable and constructive comments posted by you and the reviewers on the manuscript (Manuscript ID: ijms-2851286). We have considered all of the comments carefully and made improvement accordingly. We hope the revised manuscript will be acceptable for publication on International Journal of Molecular Science.

Best regards.

Xiangnan Li

Response to reviewer comments:

Reviewers 2:

Response to overall comments: Thanks for the comments, the manuscript has been improved accordingly.

Comment 1: Why did the authors use *rational* in the title?

Response 1: Thank you for the kind comments. We considered that nanobubble irrigation was relatively efficient and environmentally friendly in reducing the stress of ZnO-NPs on wheat. So, “rational” was been used. The title is confusing and we have changed it to “An innovative approach to alleviate zinc oxide nanoparticle stress on wheat through nanobubble irrigation”.

Comment 2: It would be nice to show the experimental system visually so that the readers get interested

Response 2: We greatly appreciate your suggestions. The experimental system has been shown in Fig.8 in the new version.

Comment 3: The title should be revised completely since the authors provide nanobubble as a solution against Zinc Oxide Nanoparticle toxicity.

Response 3: Thank you for the kind comments. We changed this title to “An innovative approach to alleviate zinc oxide nanoparticle stress on wheat through nanobubble irrigation”.

Comment 4: How did the authors choose 500 mg 569 L−1 ZnO-NPs? Were there any preliminary experiments?   

Response 4: Thank you for the kind comments. The selected 500 mg L-1 ZnO-NPs were based on our previous work by Guo et al. [21]. We have referenced the work in L60 and L566 in the new version.

21   Guo, J.; Li, S.; Brestic, M.; Li, N.; Zhang, P.; Liu, L.; Li, X. Modulations in Protein Phosphorylation Explain the Physiological Responses of Barley (Hordeum vulgare) to Nanoplastics and ZnO Nanoparticles. J Hazard Mater. 2023, 443, 130196.

Comment 5: Line 670: It is not enough to provide the reference. It is mandatory to explain the applied methods for all the parameters.

Response 5: Thank you for the kind comments. We have added some sentences about the applied methods. (L669-670, L674-678)

Comment 6: Line 704: How can the authors have concluded about the inhibition of glycolysis? Did they measure all the relevant metabolites related to glycolysis and adjacent processes?

Response 6: We greatly appreciate your suggestions. We did not measure all relevant metabolites related to glycolysis and adjacent processes. However, in our study, as compared to the water irrigation control, the conventional ZnO-NPs irrigation decreased the glucose concentration and changed the key enzyme activities related to glycolysis and pentose phosphate pathway, among which it is also noteworthy to point out that AGPase activity was increased, while PFK activity were decreased (Figure 4 A, B). AGPase is responsible for the conversion between glucose-1-phosphate and ADP-glucose, which is a limiting enzyme of starch biosynthesis [58]; PFK catalyzes an irreversible reaction in glycolysis, namely the conversion of fructose-6-phosphate to fructose-1,6-bisphosphate [59]. Hence, above changes illustrated that the conventional ZnO-NPs irrigation hindered the glycolysis in roots.

Meanwhile, most key enzyme activities related to glycolysis appeared a decreased trend in leaves (Figure 4B). In combination with the decreased tendency for most enzyme activities and glucose concentration, our results showed that glycolysis was restrained in leaves.

Reference:

58   Martin, C.; Smith, A.M. Starch Biosynthesis. Plant Cell. 1995, 7, 971-985.

59   Jian, S.; Li, S.; Liu, F.; Liu, S.; Gong, L.; Jiang, Y.; Li, X. Elevated Atmospheric CO2 Concentration Changes the Eco-Physiological Response of Barley to Polystyrene Nanoplastics. Chem Eng J. 2023, 457, 141135.

Comment 7: Authors did a very good job in presenting their study however, there are some major concerns in the title and explanation of the experimental design. Therefore, I strongly suggest the authors improve them. 

Response 7: We greatly appreciate the insightful comments and suggestions for our manuscript. We revised our manuscript according to those comments and suggestions.
